

# Enhancement of specific T-lymphocyte responses by monocyte-derived dendritic cells pulsed with E2 protein of human papillomavirus 16 and human p16INK4A

Nuchsupha Sunthamala[1,2], Neeranuch Sankla[1],
Jureeporn Chuerduangphui[2,3], Piyawut Swangphon[2,4],
Wanchareeporn Boontun[1], Supakpong Ngaochaiyaphum[1],
Weerayut Wongjampa[2,5], Tipaya Ekalaksananan[2,5] and
Chamsai Pientong[2,5]

[1] Department of Biology, Faculty of Science, Mahasarakham University, Maha Sarakham,
Thailand
[2] HPV&EBV and Carcinogenesis Research Group, Khon Kaen University, Khon Kaen, Thailand
[3] Department of Microbiology, Faculty of Science, Kasetsart University, Bangkok, Thailand
[4] Faculty of Medical Technology, Prince of Songkla University, Hat Yai, Songkla, Thailand
[5] Department of Microbiology, Faculty of Medicine, Khon Kaen University, Khon Kaen, Thailand

Corresponding author
Chamsai Pientong, chapie@kku.ac.th

## ABSTRACT

**Introduction:** Prophylactic vaccines are already available for prevention of human papillomavirus (HPV) infection. However, we still await development of therapeutic vaccines with high efficiency for stimulating specific T lymphocytes to clear HPV infection.

**Objective:** This study investigates the potential for subunits of human p16INK4a protein and E2 protein of HPV16 to stimulate dendritic cells and enhance the specific response of T lymphocytes against HPV-infected cells.

**Methodology:** Immunogenic epitopes of HPV16 E2 and p16INK4a proteins were predicted through the common HLA class I and II alleles present in the Thai population. Then, monocyte-derived dendritic cells (MDCs) were pulsed with HPV16 E2 and/or p16INK4a protein s and their maturity assessed. MDCs pulsed with either or both of these proteins at optimal concentrations were used for activation of autologous T lymphocytes and IFN-γ production was measured for specific response function.

**Results:** HPV16 E2 and p16INK4a proteins contain various immunogenic epitopes which can be presented by antigen-presenting cells via both HLA class I and II molecules. The stimulation of MDCs with either HPV16 E2 or p16INK4a proteins increased percentages and mean fluorescence intensity (MFI) of CD83$^+$ MDCs in a dose-dependent manner. An optimum concentration of 250 ng/mL and 150 ng/mL of HPV16 E2 and p16INK4a proteins, respectively, stimulated MDCs via the MAPK pathway (confirmed by use of MAPK inhibitors). T lymphocytes could be activated by MDCs pulsed with these proteins, leading to high percentages of both CD4$^+$ IFN-γ$^+$ T lymphocytes and CD8$^+$ IFN-γ$^+$ T lymphocytes. The production of IFN-γ was higher in co-cultures containing MDCs pulsed with HPV16 E2 protein than those pulsed with p16INK4a. Interestingly, MDCs pulsed with a combination of HPV16 E2 and p16INK4a significantly increased IFN-γ production of

T lymphocytes. The IFN-γ production was inhibited by both HLA class I and II blockade, particularly in co-cultures with MDCs pulsed with a combination of HPV16 E2 and p16INK4a.

**Conclusions:** This suggests that MDCs pulsed with both proteins enhances specific response of both CD4$^+$ and CD8$^+$ T lymphocytes. This study might provide a strategy for further in vivo study of stimulation of T lymphocytes for therapy of HPV-associated cancer.

## INTRODUCTION

Among cancers of women in Southeast Asia, cervical cancer has the second-highest incidence and is a common cause of deaths in low-and middle-income countries. There are around 527,624 new cases of cervical cancer annually worldwide and 265,672 deaths. Early sexual behavior and an increase in Human papillomavirus (HPV) infections has led to increasing incidence of cervical cancer in younger women (*Shrestha et al., 2018*). HPV infection causes genetic and epigenetic changes that can promote cancer development. This virus is associated with over 90% of cervical cancer cases (*De Martel et al., 2012*).

Currently, only a prophylactic vaccine is available for HPV infection: no effective therapeutic vaccine has been developed (*Langsfeld & Laimins, 2016*; *Wieking et al., 2012*). Immunotherapy is a strategy to promote the therapeutic efficiency for HPV infection. Stimulating T lymphocytes and inducing differentiation of non-protective CD8$^+$ T lymphocytes to cytotoxic T lymphocytes (CTLs) can destroy HPV-infected cells and cancer cells. In addition, long-term CD8$^+$ T cells can recognize antigens and can release cytotoxic molecules (*Palucka & Banchereau, 2013*; *Farkona, Diamandis & Blasutig, 2016*). Several mechanisms have been used to stimulate T lymphocytes for vaccine development. One is stimulation by activated dendritic cells (DCs) (*Palucka & Banchereau, 2013*). DCs are antigen presenting cells (APCs) which present antigens to T lymphocytes association with molecules of the major histocompatibility complex (MHC) class I and II. DCs can stimulate both CD4$^+$ T lymphocytes and CD8$^+$ T lymphocytes (*Yatim & Lakkis, 2015*). Therefore, the DCs might be designed as the dendritic-based vaccine for HPV infection.

More than 180 types of HPV are known and can be divided into two groups: low-risk HPV and high-risk HPV. HPV16, a high-risk HPV, is highly associated with various cancers (*Rosales & Rosales, 2014*). Some proteins of HPV are immunogenic such as E2, E6 and E7. These are all "early" proteins, expressed in an early stage of infection. The HPV E6 and E7 proteins are oncogenic, often referred to as oncoproteins. These early proteins can stimulate the adaptive immune response, but HPV can evade the immune system by modulating immune responses (*Stanley, 2006*; *Arany, Goel & Tyring, 1995*; *Ashrafi et al., 2005*). Thus, the innate immune cells that play initial roles in stimulating the adaptive immune response are important to study. HPV infection also results in over-expression of p16INK4a via the activity of the HPV E7 oncoprotein (*Ruas & Peters, 1998*). The immune

response to host p16INK4a and HPV E2 is induced via specific CD8[+] CTLs and antibody responses to virus antigens and abnormal cells (*McLaughlin-Drubin, Park & Munger, 2013*; *Piersma et al., 2008*). Therefore, pulsing of DCs with a combination of p16INK4a and HPV16 E2 proteins might be a promising strategy for enhancing specific T-lymphocyte responses against HPV infection.

Here, we aimed to develop human p16INK4a and HPV16 E2 proteins for stimulation of DCs, a link to the adaptive immune response, which enhances T-lymphocyte responses against HPV-infected cells. The HPV16 E2 and p16INK4a proteins were expressed using a bacterial system. The purified proteins were pulsed to immature monocyte-derived dendritic cells (MDCs) and presented as antigens to autologous T lymphocytes. The efficiency of the HPV16 E2 and p16INK4a proteins was evaluated in vitro by examining their capacity to induce immature MDCs to develop into mature MDCs and to enhance numbers of IFN-γ-producing CD4[+] and CD8[+] T lymphocytes. The role of mitogen-activated protein kinase (MAPK) on activation of MDCs was determined. Specific HLA blockades were used to confirm the type of T-lymphocyte responses. This research reveals important information for development of protein-based subunit vaccines for therapy of HPV-associated cancer.

## MATERIALS AND METHODS

### Preparation and purification of HPV16 E2 and p16INK4a proteins using *Escherichia coli* BL21 system

The vectors pTrcHisA-HPV16 E2 and pTrcHisA-p16INK4a were transformed using heat shock into competent *Escherichia coli* BL21 cells and the resulting cells were used for protein expression (*Servinsky et al., 2016*; *Inoue, Nojima & Okayama, 1990*). The bacteria were cultivated for 24 h in Luria-Bertani (LB) broth containing 100 μg/ul of ampicillin at 37 °C. Expression of p16INK4a and HPV16 E2 proteins was induced by adding isopropyl thiogalactoside (Thermo Scientific, Waltham, MA, US). Proteins were extracted using B-PERR Bacteria Protein Extraction Reagent (Thermo Scientific, Waltham, MA, US). The desired proteins were purified by using HisPur Cobalt Purification Kit (Thermo Scientific, Waltham, MA, US) and removed the endotoxin by using Pierce High Capacity Endotoxin Removal Resin (Thermo Scientific, Waltham, MA, US). Protein concentration was measured using Bradford's method (*Kruger, 1994*). The identities of the purified proteins were confirmed by SDS-PAGE and western blot analysis using anti-histidine antibody, anti-HPV16 E2 antibody, and anti-p16INK4A antibody (*Li et al., 2017*).

### Immunogenic epitope prediction of HPV16 E2 and p16INK4a proteins for HLA class I and II binding

The common HLA class I and II molecules in the Thai population were identified at http://www.allelefrequencies.net/. Possible binding epitopes of HLA class I and II for HPV16 E2 and p16INK4a proteins were predicted using two databases, SYFPEITHI and NetMHCpan version 4.0 (*Nielsen et al., 2010*; *Nielsen & Andreatta, 2016*). The prediction was performed using 9-meric amino-acid residues for HLA class I and 15-meric amino-acid residues for HLA class II epitopes, respectively.

### Isolation of CD14⁺ peripheral blood mononuclear cell and cultivation of MDCs

Peripheral blood was collected from five healthy volunteers who had given written consent. The study was approved by the Human Ethics Research Committee of Khon Kaen University (No. HE611307). The peripheral blood mononuclear cell (PBMC) were separated by centrifugation with Lymphoprep (Stemcell technologies, Oslo, Norway, Europe) (*Müller et al., 1993*). CD14⁺ monocytes were isolated from PBMC with CD14 magnetic particles to which an anti-human CD14 monoclonal antibody was conjugated and using the IMag separator (BD Biosciences, San Jose, CA, USA). The 95% purity of CD14⁺ monocytes was confirmed using flow cytometry. The CD14⁺ monocytes of each volunteer were separately cultured in RPMI-1640 medium (Gibco, Thermo Fisher, Waltham, MA, USA) supplemented with 10% fetal bovine serum (FBS) (Gibco, Thermo Fisher, Waltham, MA, USA), 100 U/ml recombinant human GM-CSF (Peprotech, Rocky Hill, NJ, USA), 50 U/ml recombinant human IL-4 (Peprotech, Rocky Hill, NJ, USA), 100 U/mL penicillin and 100 μg/mL streptomycin for 5–6 days (*Murray et al., 2015*; *Steevels & Meyaard, 2011*; *Xiao et al., 2018*). This cultivation method enhanced the differentiation of monocytes into immature MDCs. The CD14⁻ cells remaining after CD14⁺ isolation were cultured in RPMI-1640 supplemented with 5% FBS, 5% autologous serum, recombinant human IL-2 (20 IU/mL) (Peprotech, Rocky Hill, NJ, USA), 100 U/mL penicillin and 100 μg/mL streptomycin for T-lymphocyte proliferation (*Xiao et al., 2018*).

### HPV16 E2 and p16INK4a proteins stimulate immature MDCs

The immature MDCs were seeded into 96-well plates and stimulated to become mature MDCs by using HPV16 E2 or p16INK4a at various concentrations; 0, 50, 100, 150, 250 and 500 ng/mL (added on day 5 and incubated for 48 h). The stimulated MDCs were harvested, washed and stained with FITC-HLA-DR, PE-CD80, APC-CD86 and PE-Cy7-CD83-tagged monoclonal antibodies (BD Pharmingen, San Jose, CA, USA) and analyzed using flow cytometry (FACScanto II, BD Biosciences, San Jose, CA, USA) (*Xiao et al., 2018*).

### Evaluation the role of MAPK pathways on maturation and activation of MDCs

The role of MAPK pathways in the maturation of MDCs was determined by adding three specific kinase inhibitors into immature MDCs at day 5. The specific kinase inhibitors were p38 MAPK inhibitor (SB203580, 50 μM), ERK inhibitor (PD98059, 50 μM), and JNK inhibitor (SP600125, 20 μM). A total of 2 h after addition of these inhibitors, HPV16 E2 and/or p16INK4a were added and culture was continued for a further 48 h. The cells were collected and examined for CD83 expression by flow cytometry. The secretion of IL-12 and IL-1β was determined by ELISA (Peprotech, Rocky Hill, NJ, USA) (*Xiao et al., 2018*).

### Specific protein-pulsed MDCs stimulate autologous T-lymphocyte proliferation and IFN-γ secretion

Immature MDCs were seeded into 96-well U-bottom plates and cultured in RPMI-1640 containing 10% FBS, 100 U/ml rhGM-CSF, 50 U/ml rhIL-4, 100 U/mL penicillin, and

100 µg/mL streptomycin. The immature MDCs were further stimulated with HPV16 E2, p16INK4a, or HPV16 E2/p16INK4a at the determined optimal concentration for 24 h. The protein-pulsed MDCs were washed and co-cultured with autologous T lymphocytes in the ratio of protein-pulsed MDCs:T cells at 1:1, 1:5 and 1:10. The supernatants were harvested after 24 h of co-culture for determination of IFN-γ secretion by ELISA. The cells were harvested for staining with FITC-CD3, APC-CD4, and PerCP-Cy5.5-CD8-tagged monoclonal antibodies (BD Pharmingen, San Jose, CA, USA) and intracellular IFN-γ monoclonal antibodies (BD Pharmingen, San Jose, CA, USA) and detected using flow cytometry (*Xiao et al., 2018*).

### Evaluation of HLA blockade on stimulation of T lymphocytes by protein-pulsed MDCs

Immature MDCs were seeded into 96-well U-bottom plates and cultured in RPMI-1640 containing 10% FBS, 100 U/ml rhGM-CSF, 50 U/ml rhIL-4, 100 U/mL penicillin and 100 µg/mL streptomycin. The immature MDCs were further stimulated with HPV16 E2, p16INK4a, or HPV16 E2/p16INK4a at the determined optimal concentration for 24 h. Then, protein-pulsed MDCs were washed and blockade antibodies added. These were mouse anti-human HLA-ABC monoclonal antibody (eBioscience, San Diego, CA, USA) and mouse anti-human HLA-DR, DP, DQ monoclonal antibodies (BD Pharmingen, San Jose, CA, USA) for 2 h (*Rahman et al., 2017*). The protein-pulsed MDCs were washed and co-cultured with autologous T lymphocytes in the ratio of protein-pulsed MDCs:T cells at 1:1, 1:5 and 1:10. The supernatants were harvested after 24 h of co-culture for determination of IFN-γ secretion by ELISA. The cells were harvested for staining with FITC-CD3, APC-CD4 and PerCP-Cy5.5-CD8-tagged monoclonal antibodies (BD Pharmingen, San Jose, CA, USA) and intracellular IFN-γ monoclonal antibodies (BD Pharmingen, San Jose, CA, USA) and detected using flow cytometry (*Xiao et al., 2018*; *Rahman et al., 2017*).

### Statistical analysis

Experiments were performed independently three times. The data were analyzed using analysis of variance and differences among the experimental groups tested using Tukey's multiple-comparison test implemented in Prism version 5. The results are shown as the mean and standard deviation (SD). The symbols * and # indicate statistical significance, at $p < 0.05$, in comparisons with controls or between groups, respectively.

## RESULTS

### Frequency of common HLA class I and II alleles in the Thai population and prediction of the immunogenic epitopes of HPV16 E2 and p16INK4a proteins

Online database information about common HLA class I and class II alleles in Thailand was based on data from 16,807 people. The three most common HLA class I alleles were HLA-A*02, HLA-A*11 and HLA-A*24 with frequencies of 29.2, 27.7 and 17.3%, respectively. The three most common HLA class II alleles were HLA-DRB1*15,

Table 1 The table showed % frequency of HLA class I and II alleles that are commonly found in the Thai population.

| HLA class I | | HLA class II | |
|---|---|---|---|
| Allele | % Allele frequency | Allele | % Allele frequency |
| HLA-A*02 | 29.2 | HLA-DRB1*04 | 14.4 |
| HLA-A*11 | 27.7 | HLA-DRB1*09 | 11.5 |
| HLA-A*24 | 17.3 | HLA-DRB1*12 | 16.9 |
| HLA-A*33 | 13.8 | HLA-DRB1*15 | 17.5 |
| HLA-B*15 | 14.8 | | |
| HLA-B*40 | 13.4 | | |
| HLA-B*46 | 13.2 | | |

HLA-DRB1*12 and HLA-DRB1*04 with frequencies of 17.5, 16.9 and 14.4%, respectively (Table 1).

The prediction of HLA class I and class II epitopes for HPV16 E2 and p16INK4a proteins was performed using two databases, the 9-meric amino-acid residues and 15-meric amino-acid residues were predicted for HLA class I and class II alleles, respectively. The HPV16 E2 protein (365 amino-acid residues) has immunogenic epitopes for both types of HLA. Fig. 1A shows common immunogenic epitopes of HPV16 E2 protein for HLA class I alleles including HLA-A*02:01, HLA-A*11:01, HLA-A*24:02, HLA-B*15:01 and HLA-B*40:01, and for HLA class II alleles including

HLA-DRBI*04:01 and HLA-DRBI*15:01. Similarly, the p16INK4a protein (156 amino-acid residues) also presents immunogenic epitopes for both types of HLA including HLA-A*02:01, HLA-A*11:01, HLA-A*24:02, HLA-B*15:01, HLA-B*40:01, HLA-DRBI*04:01 and HLA-DRBI*15:01 (Fig. 1B). This suggests that these two proteins can be presented by antigen-presenting cells via HLA class I and class II molecules.

## Production of HPV16 E2 and p16INK4a protein in *Escherichia coli* BL21

HPV16 E2 and p16INK4a proteins expressed in *E. coli* BL21 were purified and eluted twice. Identities of the proteins were confirmed by western blot using specific antibodies for polyhistidine and HPV16 E2, respectively. We found the purified p16INK4a protein with a polyhistidine tag was of the expected size, 18 kDa (Fig. 1C). The purified HPV16 E2-polyhistidine tagged protein was 43 kDa (Fig. 1D). Therefore, these proteins were further used for stimulating of MDCs and T lymphocytes.

## HPV16 E2 and p16INK4a proteins induced phenotypic maturation of MDCs

We determined the optimum concentrations of HPV16 E2 and p16INK4a proteins for stimulating the maturation of MDCs. Evaluation of the expression level of CD83, which is a marker of mature DCs, was done after stimulating CD14$^+$ monocytes with HPV16 E2 and/or p16INK4a at various concentrations; 0, 50 100, 150, 250 and 500 ng/mL (treated on day 5 and cultured for 48 h). We observed that CD14$^+$ monocytes cultured with

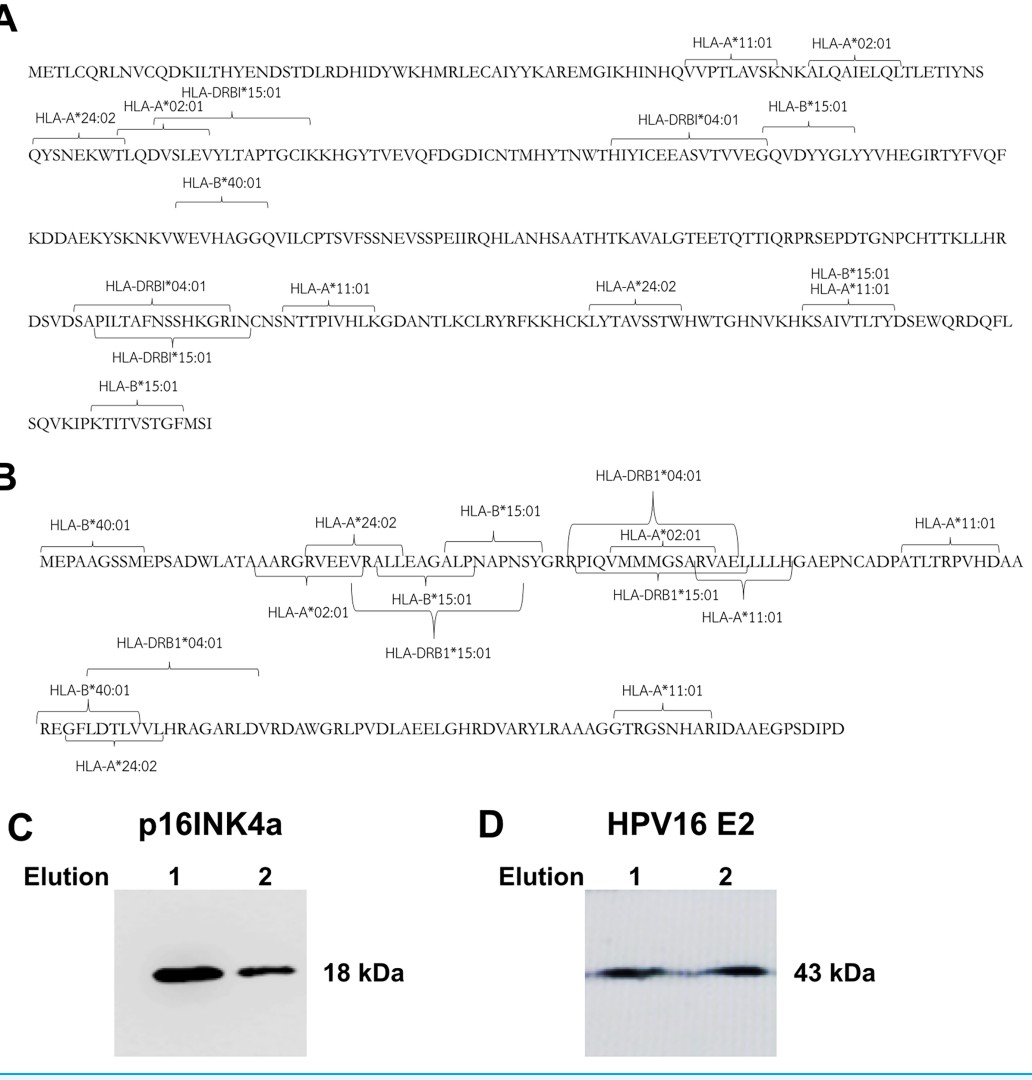

**Figure 1 Mapping of immunogenic epitopes and protein sizes of p16INK4a and HPV16 E2.** Possible immunogenic epitopes of (A) HPV16 E2 and (B) p16INK4a were mapped using two databases, SYF-PEITHI and NetMHCpan version 4.0. The prediction was performed by using 9-meric amino-acid residues for HLA class I epitope binding and 15-meric amino-acid residues for HLA class II, respectively. The protein sizes of (C) p16INK4a and (D) HPV16 E2 were checked by western blot.

rhGM-CSF and rhIL-4 can be induced to develop to immature monocyte-derived DC (Fig. 2B) when compared to negative control (CD14[+] monocytes without rhGM-CSF and rhIL-4) (Fig. 2A). The immature DCs were stimulated with HPV16 E2 and p16INK4a proteins resulting in their differentiation into mature DCs with more cytoplasmic protrusions (Figs. 2C and 2D). However, their phenotypic were not significantly observed.

Both the percentage of positive cells and mean fluorescence intensity (MFI) of CD83[+] MDCs increased in a dose-dependent manner after stimulation with HPV16 E2 protein to peak at a concentration of 250 ng/mL (Figs. 2C and 2D). For p16INK4a, percentage of positive cells and MFI of CD83[+] MDCs peaked at 150 ng/mL (Figs. 2E and 2F). Therefore, the optimum concentration of HPV16 E2 and

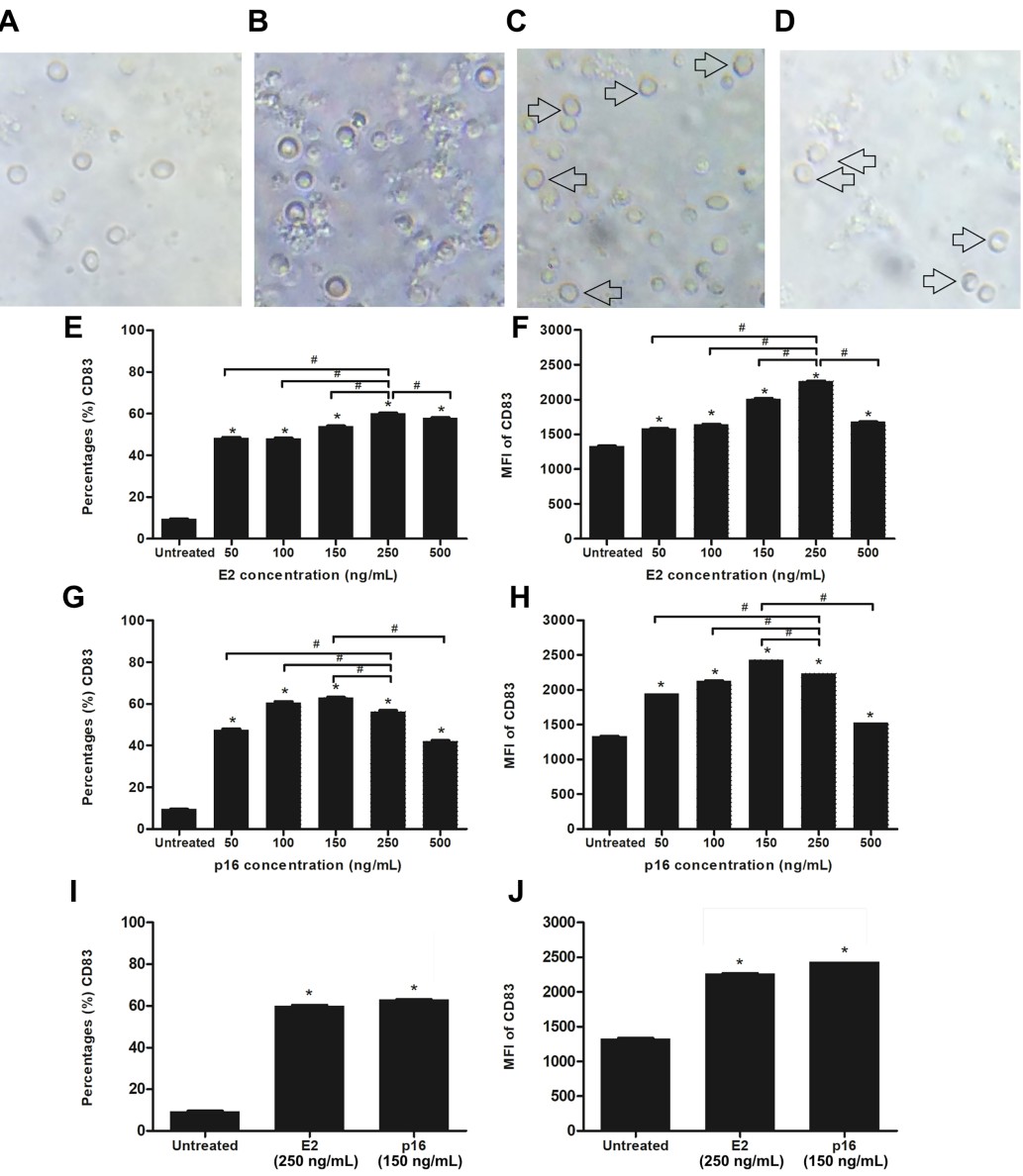

**Figure 2 Stimulation of immature MDCs by HPV16 E2 and p16INK4a proteins.** (A) Characteristics of CD14+ monocytes cultured in RPIM-1640 medium without GM-CSF and IL-4. (B) Characteristics of CD14+ monocytes in RPIM-1640 medium with GM-CSF and IL-4 before stimulation. (C) Characteristics of mature MDCs treated with HPV16E2 at concentration of 250 ng/mL. (D) Characteristics of mature MDCs treated with p16INK4A at concentration of 150 ng/mL. (E) Percentage and (F) mean fluorescence intensity (MFI) of CD83+ MDCs stimulated by HPV16 E2 protein at various concentrations. (G) Percentage and (H) MFI of CD83+ MDCs stimulated by p16INK4a protein at various concentrations. (I) Percentage and (J) mean fluorescence intensity (MFI) of CD83+ MDCs stimulated by HPV16E2 and p16INK4A proteins at best concentrations. The untreated group is immature MDCs without stimulation with any proteins. The experiments were performed by using five healthy volunteers and repeated the experiment three times. (*) $p$-value $< 0.05$ compared to control. (#) $p$-value $< 0.05$ compared between groups.

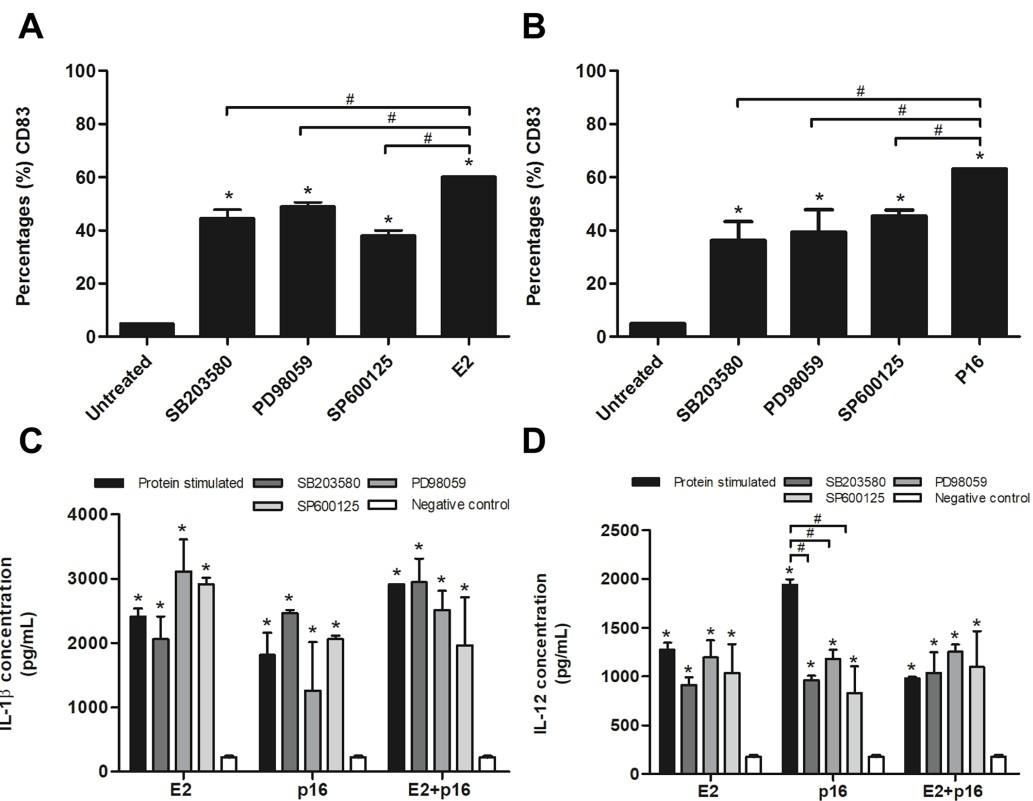

**Figure 3 Expression of CD83 and cytokine production by MDCs stimulated by HPV16 E2 and/or p16INK4a proteins.** (A) The expression of CD83 stimulated by HPV16 E2 and (B) p16INK4a after treatment with p38 MAPK inhibitor (SB203580), ERK inhibitor (PD98059), or JNK (SP600125), examined using flow cytometry. (C) The production of IL-1β (pg/mL) and (D) IL-12 after blocking of MDCs with SB203580, PD98059 or SP600125, then stimulated by HPV16 E2 and/or p16INK4a proteins. The HPV16 E2 protein concentration was 250 ng/mL and p16INK4A protein concentration was 150 ng/mL. The experiments were performed by using five healthy volunteers and repeated the experiment three times in each volunteer blood sample. (*) $p$-value $< 0.05$ compared to control. (#) $p$-value $< 0.05$ compared between groups.

p16INK4a proteins for further experiments were 250 ng/mL and 150 ng/mL, respectively (Figs. 2I and 2J).

## HPV16 E2 and p16INK4a proteins stimulate the maturation and activation of MDCs via MAPK pathway

To confirm whether HPV16 E2 and p16INK4a proteins stimulated immature MDCs via the MAPK pathway, immature MDCs were treated with p38 MAPK, ERK and JNK inhibitors in separate experiments prior to stimulation with the determined optimum concentrations of HPV16 E2 and/or p16INK4a. We found that the inhibitors significantly decreased the percentage of CD83[+] MDCs in cultures stimulated by HPV16 E2 or p16INK4a proteins (Figs. 3A and 3B). Secretion levels of IL-1β did not significantly differ between all treatment groups (Fig. 3C). However, secretion of IL-12 was significantly lower in experiments where MAPK inhibitors were administered in conjunction with p16INK4a relative to p16INK4a alone (Fig. 3D). In contrast, IL-12 levels were not

suppressed when MAPK inhibitors were administered in conjunction with HPV16E2 or HPV16 E2 together with p16INK4a (Fig. 3D). These findings suggest that the activation of MDCs by HPV16 E2 and/or p16INK4a proteins is via the MAPK pathway.

## MDCs pulsed with HPV16 E2 and p16INK4a stimulate CD4$^+$ and CD8$^+$ T lymphocytes to produce IFN-$\gamma$

To determine the capability of MDCs to process immunogenic epitopes, the populations of CD4$^+$ and CD8$^+$ T lymphocytes were examined. MDCs pulsed with HPV16 E2 and/or p16INK4a were co-cultured with IL-2-cultured T lymphocytes in ratios of 1:1, 1:5 and 1:10. Percentages of CD4$^+$ and CD8$^+$ T lymphocytes were not significantly different in all ratios (Figs. 4A and 4B). Interestingly, the percentage of CD4$^+$ IFN-$\gamma^+$ T lymphocytes dramatically increased ratio-wise after co-culture with MDCs pulsed with HPV16 E2 and/or p16INK4a (Fig. 4C). The percentage of CD8$^+$ IFN-$\gamma^+$ T lymphocytes did not significantly differ between treatments at ratios of 1:5 and 1:10. However, at the ratio of 1:1, the percentage of these lymphocytes was significantly greater when MDCs were pulsed with HPV16 E2 or p16INK4a relative to the combination of HPV16 E2 and p16INK4a (Fig. 4D). This indicates that HPV16 E2 might exhibit higher immunogenicity than the combination of HPV16 E2 and p16INK4a. The activation of these proteins can be triggered via CD4$^+$ or CD8$^+$ lymphocytes. We further confirmed the secretion of IFN-$\gamma$ using ELISA. Production of IFN-$\gamma$ was higher in co-cultures where MDCs had been pulsed with HPV16 E2 rather than p16INK4a, and varied according to the ratio. The combination of HPV16 E2 and p16INK4a significantly increased the production of IFN-$\gamma$ at ratios of 1:5 and 1:10, while HPV16 E2 alone strongly induces the production of IFN-$\gamma$ in a ratio-dependent manner (Fig. 4E). This suggests that the activation of T lymphocytes by HPV16 E2 and p16INK4a for IFN-$\gamma$ production might be processed by CD4$^+$ and CD8$^+$ T lymphocytes.

## CD4$^+$ and CD8$^+$ T lymphocytes co-response to combination of HPV16 E2 and p16INK4a pulsed MDCs

To determine the effectiveness of HPV16 E2 and/or p16INK4a in stimulation of CD4$^+$ or CD8$^+$ T lymphocytes, HLA blockade antibodies were specifically applied in co-culture conditions at a ratio of 1:5. We found inhibition of IFN-$\gamma$ production in both HLA class I and II blockade in co-culture with MDCs pulsed with a combination of HPV16 E2 and p16INK4a (Fig. 4F). This effect was not apparent when the MDCs were pulsed with either HPV16 E2 alone or p16INK4a alone (Fig. 4F). Suggesting that the stimulation with combination of HPV16 E2 and p16INK4a clearly enhances response of both CD4$^+$ and CD8$^+$ T lymphocytes.

## DISCUSSION

Administration of protein subunits is a safe and effective strategy for HPV therapy by stimulating immune responses (Li et al., 2017). The purpose of vaccine therapy against tumor or viral infection is to stimulate T lymphocytes and differentiation of non-protective cells into functional CTLs. The stimulation of T lymphocytes through DCs, professional

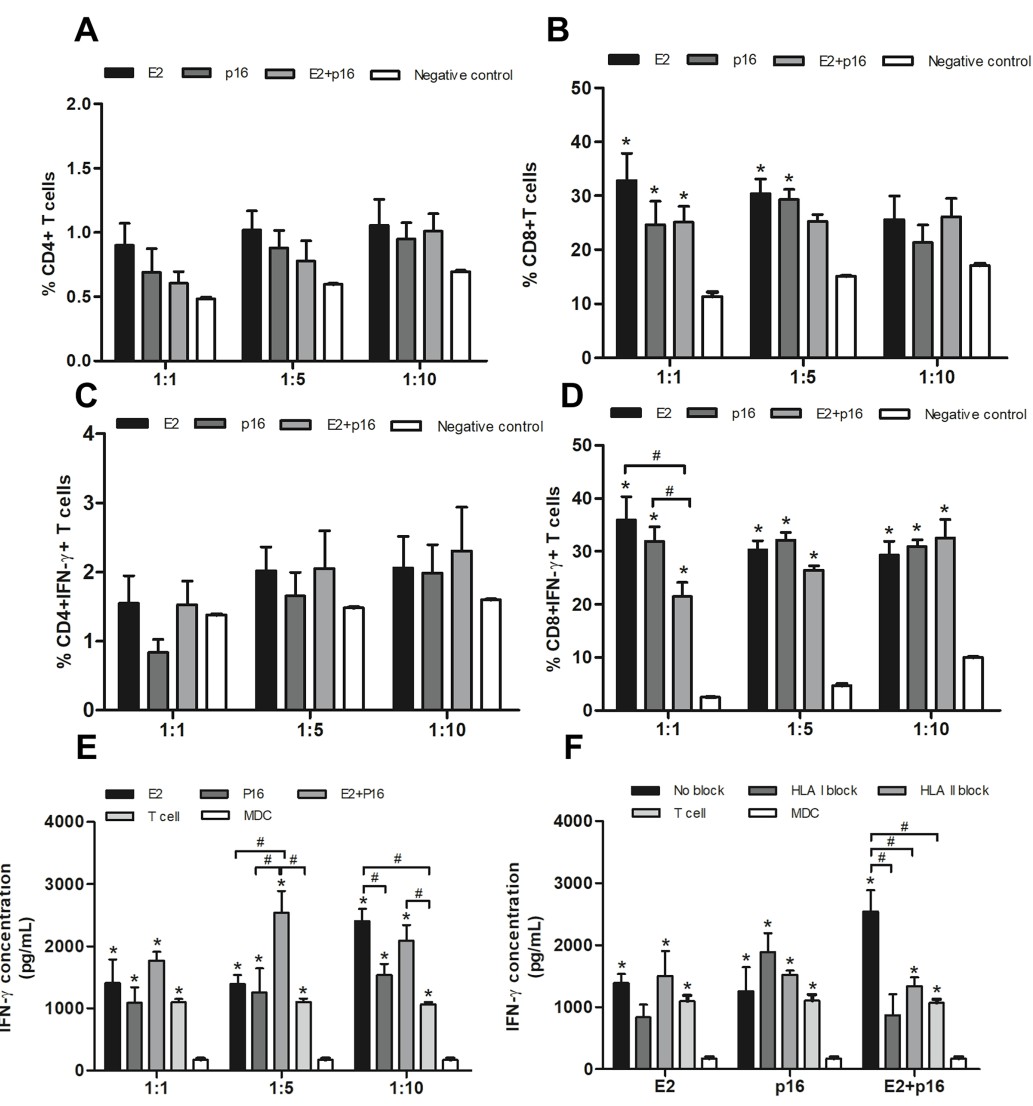

**Figure 4 MDCs pulsed with HPV16 E2 and/or p16INK4a stimulate CD4+ and CD8+ T lymphocytes.** (A) The percentage of CD4+ T lymphocytes and (B) CD8+ T lymphocytes in co-cultures with MDCs pulsed with HPV16 E2 and/or p16INK4a at various ratios. (C) The percentage of IFN-γ+CD4+ T lymphocytes and (D) IFN-γ+CD8+ T lymphocytes in co-cultures with MDCs pulsed with HPV16 E2 and/or p16INK4a at various ratios. (E) The concentration of IFN-γ (pg/mL) after co-culture of T lymphocytes with MDCs pulsed with HPV16 E2 and/or p16INK4a at various ratios. (F) The concentration of IFN-γ (pg/mL) after blockade with HLA class I and II antibodies and co-culture of T lymphocytes and MDCs pulsed with HPV16 E2 and/or p16INK4a at a ratio of 1:5. The HPV16 E2 protein concentration was 250 ng/mL and p16INK4A protein concentration was 150 ng/mL. The experiments were performed by using five healthy volunteers and repeated the experiment three times in each volunteer blood sample. (*) $p$-value $< 0.05$ compared to control. (#) $p$-value $< 0.05$ compared between groups.

APCs, is an efficient method (*Palucka & Banchereau, 2013*). Presentation of antigens by APCs to T lymphocytes occurs through the HLA molecule: the HLA class II-bound peptide antigen is presented to CD4+ T lymphocytes whereas the HLA class I-bound peptide antigen is presented to CD8+ T lymphocytes (*Yatim & Lakkis, 2015*). The HPV16 E7 protein is a well-known antigen but has relatively low CTL epitopes which can only be

presented in association with HLA-A*33:03, an HLA class I allele (*Kim et al., 2017*). The HPV16 E6 protein has a variety of epitopes that, in Chinese adolescents, can be presented in association with HLA class II molecules such as HLA-DRB1 and HLA-DQB1 (*Hu et al., 2017*). A number of HLA alleles of both classes are common in Thailand (Table 1) and both HPV16 E2 and p16INK4a proteins exhibit multiple epitopes that can be recognized by these alleles (Figs. 1A and 1B). HPV16 E2 and p16INK4a proteins are strong targets for stimulation of T lymphocytes via DCs.

The protein subunit was prepared using a popular bacterial system, *E. coli* BL21. This bacterial system capable provides the protein expression by regulating via T7 promoters which also used to produce the expression of HPV16 L1 (*Bang et al., 2015*) and HPV E7 (*Viana et al., 2009*). Western blot indicated the size of our HPV16 E2-polyhistidine tagged protein to be 43 kDa (Fig. 1C) a size comparable to that reported previously for HPV16 E2-GFP (62 kD, but including the GFP of about 27 kDa) (*Sunthamala et al., 2014*). The size of our p16INK4a-polyhistidine tagged protein was 18 kDa (Fig. 1D), consistent with other reports.

The stimulation of DCs by MtHSP70-FPR1 fusion protein, or Human Immunodeficiency Virus (HIV)-1, can induce maturation of these by increasing the expression of CD83, a marker for mature DCs, and secretion of IL-1β and IL-12 (*Xiao et al., 2018*; *Wilflingseder et al., 2004*). We found that the HPV16 E2 and p16INK4a proteins also increase the expression of CD83 in MDCs in a dose-dependent manner and induce cytoplasmic protrusions (Fig. 2). Mature DCs play an important role as APCs to stimulate the differentiation of naïve T lymphocytes into functional CTLs. However, the results of other MDCs maturation markers, including CD80, CD86 and HLA-DR that were not significantly up-regulated in the experiment (https://figshare.com/articles/HPV16E2_and_p16INK4A_project/1149351) as well as other costimulating molecule are required for T cells response so these might be changed by the HPV protein treatment of DCs (*Palucka & Banchereau, 2013*). The maturation of DCs also increases their secretion of IL-1β and IL-12, which are important in initiating T-lymphocyte responses and enhancing the immune response. IL-12 stimulates the production of IFN-γ by T lymphocytes whereas IL-1β can stimulate the growth of MDCs (*Xiao et al., 2018*). The MAPK-ERK signaling pathway controls the survival of DCs which can be stimulated by lipopolysaccharide (*Rescigno et al., 1998*). We found that HPV16 E2 and p16INK4a proteins stimulate the secretion of IL-1β and IL-12 while the partially suppression was found only in p16INK4A stimulation of IL-12 secretion by MAPK inhibitors (Figs. 3C and 3D). Interestingly, the MAPK inhibitors significantly suppress the expression of CD83 in DCs pulsed with HPV16 E2 or p16INK4a (Figs. 3A and 3B). These results suggest that the subunit proteins, HPV16 E2 and p16INK4a, individually or in combination, stimulate the maturation of MDCs thus enhancing their APC properties.

Cytotoxic T lymphocytes play an important role in cancer treatment based on immune responses. Among other approaches, CTLs can be stimulated by tumor-specific antigens or tumor-associated antigens. Tumor-specific antigens in the case of HPV infection include the non-structural proteins HPV16 E7 and HPV18 E7, which can stimulate specific T lymphocytes via APCs (*Ferrara et al., 2003*). The structural L1 protein

of HPV16 and HPV18 has been used as prophylactic vaccine against HPV in combination with adjuvant system 04 and also stimulates the maturation of DCs, secretion of pro-inflammatory cytokines and stimulation of cytotoxic activity against HPV-infected tumors (*Van den Bergh et al., 2014*). There have also been reports about the use of a combination of non-structural HPV16 E7 protein and PD-L1 blockade, an immune blockade, which exhibited highly effective stimulation of DCs in vivo and an effective CTL response (*Liu et al., 2016*). A modified vaccinia virus, Ankara virus encoding bovine papillomavirus E2 (MVA-E2), enhances the elimination of CIN lesions in the cervix (*Rosales et al., 2014*). The MtHSP70-FPR1 fusion protein, a tumor-associated antigen has recently been reported to stimulate DCs through the p38 MAPK signaling pathway and to promote the specific responses of CTLs to cervical cancer cells (*Xiao et al., 2018*). Moreover, a combination of tumor-specific antigens and tumor-associated antigens may also stimulate immune response. An example is the DNA encoding HPV-16 E7/ HSP70 fusion protein, which can stimulate the specific antitumor responses in mice (*Soleimanjahi et al., 2017*). The fusion protein HPV-16 E6/E7 and Fms-like tyrosine kinase-3 ligand also stimulates the response of CD8$^+$ T cells specific to E6 and E7 (*Li et al., 2017*). However, a strategy to develop a therapeutic vaccine against HPV infection is not well developed.

The tumor-specific antigen, HPV E2, is an early protein that controls the expression of E6 and E7 oncoproteins and is more immunogenic than either of these oncoproteins (*Pérez-Plasencia, Dueñas-Gonzalez & Bustos-Martínez, 2008*). The tumor-associated antigen, human p16INK4a, is a protein that acts as a tumor suppressor and is over-expressed in HPV infection (*Lechner et al., 2018*). Therefore, we considered HPV16 E2 and p16INK4a to be important and interesting target proteins to stimulate the immune system of patients infected with HPV. In this study, MDCs pulsed with HPV16 E2 and/or p16INK4a were able to stimulate autologous T lymphocytes to proliferate and to secrete IFN-γ. The IFN-γ is produced by CD8$^+$ T lymphocytes that eliminate infected cells and CD4$^+$ T lymphocytes that promote cell responses through the presentation of HLA class I and II molecules (*Palucka & Banchereau, 2013*; *Yatim & Lakkis, 2015*). Particularly, the combination of HPV16 E2 and p16INK4a proteins can promote greater T lymphocyte responses either protein alone, which obviously stimulates via HLA class I and II (Figs. 4E and 4F). The optimum concentrations of the two proteins in combination, as used in this study to activate MDCs, were 250 ng/ml for HPV16 E2 and 150 ng/ml for p16INK4a. Combined subunit vaccines generally use different concentrations of each antigen to elicit effective immune responses. An example is the combined protein vaccine for diphtheria-tetanus-pertussis (*Centers for Disease Control and Prevention, 2018*; *World Health Organization, 2014*). Our study might provide a strategy for development of HPV infection treatment using the optimum concentrations of HPV16 E2 and p16INK4a proteins for stimulation of T lymphocytes.

## CONCLUSIONS

In this study, we have evaluated immune-cell responses to HPV16 E2 and p16INK4a proteins in blood from healthy volunteers. HPV E2 is a tumor-specific antigen and human

p16INK4a is a tumor-associated antigen that can stimulate MDCs cells through MAPK pathways and increase the autologous T-lymphocyte response. The combination of HPV16 E2 and p16INK4a proteins increases specific responses in both CD4$^+$ and CD8$^+$ T lymphocytes. Both proteins can stimulate cell proliferation and IFN-$\gamma$ secretion by CD8$^+$ T lymphocytes (that eliminate infected cells) and CD4$^+$ T lymphocytes (that promote cell responses) through the presentation of HLA class I and II molecules, respectively. We determined that the optimum concentration of the proteins in combination was 250 ng/ml of HPV16 E2 and 150 ng/ml of p16INK4a. This study provides strategies and optimal concentrations for developing therapies to treat HPV infections using a combination of HPV16 E2 and p16INK4a proteins for stimulation of T lymphocytes.

## ACKNOWLEDGEMENTS

We sincerely thank research team members of Mahasarakham University and Khon Kean University for helpful discussions. We would like to acknowledge Prof. David Blair, for editing the MS via Publication Clinic KKU, Thailand.

### Funding

This work was performed using financial support from Khon Kean University (grant numbers 61004603 and 620008005) and scholarships under the Post-doctoral Program from Research Affairs and Graduate School, Khon Kaen University (grant no. 58222 and PD2562-12). The funders had no role in study design, data collection and analysis, decision to publish, or preparation of the manuscript.

### Grant Disclosures

The following grant information was disclosed by the authors:
Khon Kean University: 61004603 and 620008005.
Research Affairs and Graduate School, Khon Kaen University: 58222 and PD2562-12.

### Competing Interests

The authors declare that they have no competing interests.

### Author Contributions

- Nuchsupha Sunthamala conceived and designed the experiments, performed the experiments, analyzed the data, prepared figures and/or tables, authored or reviewed drafts of the paper, and approved the final draft.
- Neeranuch Sankla performed the experiments, analyzed the data, prepared figures and/or tables, authored or reviewed drafts of the paper, and approved the final draft.
- Jureeporn Chuerduangphui performed the experiments, authored or reviewed drafts of the paper, and approved the final draft.
- Piyawut Swangphon conceived and designed the experiments, analyzed the data, authored or reviewed drafts of the paper, and approved the final draft.

- Wanchareeporn Boontun performed the experiments, authored or reviewed drafts of the paper, and approved the final draft.
- Supakpong Ngaochaiyaphum performed the experiments, authored or reviewed drafts of the paper, and approved the final draft.
- Weerayut Wongjampa performed the experiments, prepared figures and/or tables, authored or reviewed drafts of the paper, and approved the final draft.
- Tipaya Ekalaksananan conceived and designed the experiments, analyzed the data, prepared figures and/or tables, authored or reviewed drafts of the paper, and approved the final draft.
- Chamsai Pientong conceived and designed the experiments, analyzed the data, prepared figures and/or tables, authored or reviewed drafts of the paper, and approved the final draft.

## Human Ethics

The following information was supplied relating to ethical approvals (i.e., approving body and any reference numbers):

The study was approved by the Human Ethics Research Committee of Khon Kaen University (No. HE611307).

## Data Availability

The raw data is available at Figshare:

Sunthamala, Nuchsupha (2020): HPV16E2 and p16INK4A project. figshare. Figure. DOI 10.6084/m9.figshare.11493516.v1.

## Supplemental Information

Supplemental information for this article can be found online at http://dx.doi.org/10.7717/peerj.9213#supplemental-information.

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
