# Peer review of "Enhancement of specific T-lymphocyte responses by monocyte-derived dendritic cells pulsed with E2 protein of human papillomavirus 16 and human p16INK4A"

_PeerJ, doi:10.7717/peerj.9213_

## Round 0.1 · original submission · Major Revisions

· Academic Editor

Major Revisions

Please address critiques of all reviewers and revise your manuscript accordingly

·

Basic reporting

No comment.

Experimental design

All experiments should have correct control groups.

Validity of the findings

No comment.

Additional comments

High-risk human papillomavirus (HPV) infection is known to be a necessary factor for cervical and anogenital malignancies. Cervical cancers account for over a quarter of a million deaths annually. Despite the availability of prophylactic vaccines, HPV infections remain extremely common worldwide. Furthermore, these vaccines are ineffective at clearing pre-existing infections and associated preinvasive lesions. As cervical dysplasia can regress spontaneously, a therapeutic HPV vaccine that boosts host immunity could have a significant impact on the morbidity and mortality associated with HPV. In this manuscript, Dr. Sunthamala and colleagues looked at the role of the HPV16 E2 and p16INK4a protein in stimulation of T lymphocytes, and wanted to provide a strategy for further in vivo study of stimulation of T lymphocytes for therapy of HPV-associated cancer. The study provides the conclusion, which showed that monocyte-derived dendritic cells pulsed with both proteins enhances specific response of both CD4+ and CD8+ T lymphocytes. Although, the current study is interesting but there are some major concerns that need to be addressed.

Major concerns:
1. “Materials and methods – Isolation of CD14+ peripheral blood mononuclear cell (PBMC) and cultivation of monocyte-derived dendritic cells (MDCs)” Line 131. Were each volunteers’ PBMC separated or mixed for subsequent experiments? Please state clearly.
2. “Results – HPV16 E2 and p16INK4a proteins induced phenotypic maturation of MDCs” Line 230-231, Figure 2A. “We observed that CD14+ monocytes cultured with rhGM-CSF and rhIL-4 (Fig 2A) can be induced to develop from immature to mature dendritic morphology.” However, there were characteristics of immature MDCs before stimulation in figure 2A!
3. “Results – HPV16 E2 and p16INK4a proteins induced phenotypic maturation of MDCs” Line 226-233, Figure 2A and B. It will be better to present the photos of before and after treatment in four groups, which were negative control (PBS or something), positive control (rhGM-CSF and rhIL-4 treatment), HVP16 E2 treatment (best condition, might 250 ng/ml) and p16INK4A treatment (best condition, might 150 ng/ml).
4. “Results – HPV16 E2 and p16INK4a proteins induced phenotypic maturation of MDCs” Line 234-239, Figure 2C to F. It is necessary to present the positive control (rhGM-CSF and rhIL-4 treatment) in all charts, and it will be better using “Negative control” than “Untreated”.
5. “Results – HPV16 E2 and p16INK4a proteins induced phenotypic maturation of MDCs” Line 226-239, Figure 2. To make the evidence stronger, it will be better to present the expression level of CD83 before and after treatment with HPV16 E2 (best condition, might 250 ng/ml) and p16INK4A (best condition, might 150 ng/ml) respectively in more cases of mature DCs, which separate from 5-8 other volunteers respectively.
6. “Materials and methods – Evaluation the role of MAPK pathways on maturation and activation of MDCs” Line 155-158. “The specific kinase inhibitors were p38 MAPK inhibitor (SB203580, 50 μM), ERK inhibitor (PD98059, 50 μM), and JNK inhibitor (SP600125, 20 μM).” Why use the concentration of these inhibitors? Please state clearly and provide references.
7. “Results – HPV16 E2 and p16INK4a proteins stimulate the maturation and activation of MDCs via MAPK pathway” Line 248-251, Figure 3C and D. It is necessary to present the levels of IL-1β and IL-12 before and after treatment with HVP16 E2 (best condition, might 250 ng/ml) and p16INK4A (best condition, might 150 ng/ml) respectively.
8. “Discussion” Line 317-319. “We found that HPV16 E2 and p16INK4a proteins stimulate the secretion of IL-1β and IL-12 which was partially suppressed by MAPK inhibitors (Figs 3C and 3D).” I did not see any data shown that PV16 E2 and p16INK4a proteins stimulate the secretion of IL-1β and IL-12.
9. “Results – MDCs pulsed with HPV16 E2 and p16INK4a stimulate CD4+ and CD8+ T lymphocytes to produce IFN-γ” Line 256-260, Figure 4A and B. It is necessary to present the populations of CD4+ and CD8+ T lymphocytes with negative control (non-stimulating the maturation of MDCs). It also need negative control in Figure 4C and D.
10. “Results – MDCs pulsed with HPV16 E2 and p16INK4a stimulate CD4+ and CD8+ T lymphocytes to produce IFN-γ” Line 265-266. “This indicates that HPV16 E2 might exhibit higher immunogenicity than the combination of HPV16 E2 and p16INK4a.” To make the evidence stronger, it will be better to present the experiment in more cases of mature DCs, which separate from 5-8 other volunteers respectively.
11. “Results – CD4+ and CD8+ T lymphocytes co-response to combination of HPV16 E2 and p16INK4a pulsed MDCs” Line 265-266. “Suggesting that the stimulation with combination of HPV16 E2 and p16INK4a clearly enhances response of both CD4+ and CD8+ T lymphocytes.” To make the evidence stronger, it will be better to present the experiment in more cases of mature DCs, which separate from 5-8 other volunteers respectively.

Minor comments:
1. The ABSTRACT need to be rewrite. It does not need to contain much background information in the ABSTRACT section.
2. Figure 3C and D. What is the concentration of HPV16 E2 and p16INK4a in “E2+p16” groups?
2. Figure 4. What is the concentration of HPV16 E2 and p16INK4a in “E2+p16” groups?

Reviewer 2 ·

Basic reporting

The abstract should be structured.

The introduction is too long, especially the last paragraph, where the authors should shortly describe the general aim. Some technical aspects mentioned here would be fit better the Materials and Methods section, maybe before the subsections describing the specificities.

Experimental design

Methods are overall clear and the experimental procedures are reasonable. The authors declared they used 5 healthy volunteers to get the blood, but they repeated the experiment three times: each time did they make the experiments with the purified cells from all of them?
By the way, did the authors assess the HLA genotype of the healthy donors?

Validity of the findings

Again, from the figures and legends, it is not clear how the cells of the 5 healthy volunteers have been used.

Additional comments

See above comments. In general, the manuscript is clearly written. A few experimental details should be clarified.

Reviewer 3 ·

Basic reporting

Dendritic cells (DCs) are potent antigen-presenting cells of the immune system and their maturation status is critical in terms of both cytokine secretion and stimulating resting T cells. Several in vitro approaches to stimulate DCs maturation pulsed though foreign antigenic proteins has been tested and employed in T cells activation. In this manuscript, author investigated that human PBMCs stimulated with tumor associated protein HPV E2 and p16INK4a differentiated into mature DCs leading to T cell activation. This manuscript is well conceptualized and written, however limited experimentation to confirm integrity of purified proteins and lack of any data that could indicate the maturating DCs are not contaminated with other cells types weaken the effectiveness of this study.

Experimental design

Authors need to provide some strong biochemical and biophysical characterization of the purified protein, given that both the proteins are not of bacterial origin and chances of them to form inclusion bodies are very high that leads to synthesis of immature proteins. Immune signaling from NF-kB pathways drives secretion of major inflammatory mediators, nothing described from NF-kB pathway further diluting the outcome of this study. Additionally, it is required to monitor the surface markers for CD3, CD19 and CD14 to rule out the possibility of other cell type contamination post DCs maturation. Some other points are-
1. Figure 1C & D- better to provide SDS staining with un-induced control, also in supplementary figure for uncropped WB of protein it seems that the gel background of marker lane and protein lane has different contracts settings.
2. Figure 2A & B- Post 48h of protein incubation, the status of protein integrity is a major concern in culture medium. It is not tested whether proteins have no disintegration in culture media post incubations.
3. Figure 2- what is the status of B cell and macrophages population post protein treatment.
4. Figure 3- measuring expression of total MAPK are not a good readout to confirm the induction of signaling pathways, it is better to analyze the phosphorylation status of these proteins.
5. Figure 3D in PBS lane cell exposed to combined protein treatment has less IL12, compared to individual treatments, any explanation?

Validity of the findings

Authors have analyzed that around 60% immature DCs are converted to mature DCs at the peak concentration of proteins, still 40% cells were unable to respond which can be seen in Figure 4A & B with no significant induction of CD4+ and CD8+ cells population. Moreover, authors have not provided any positive control with protein treatment to compare the outcomes from matured DCs.

Additional comments

Authors have shown the in vitro maturation of DCs through protein pulsing followed by activation of T cells population, still authors need to add some more data that will further enhance the outcomes of this manuscript.

·

Basic reporting

Manuscript is well written and figures, legends are neat. Discussion can be improved.

Experimental design

Expt design was ok but could be more rigorous with controls

Validity of the findings

please see pdf comments and word doc concerns that need to be addressed

Additional comments

Sunthamala et al use recombinant HPV proteins to induce DC maturation that they show can enhance CD4/CD8 T cell responses in the manuscript herein.
The manuscript is written well. The scope of the study is limited but still important enough to encourage science and research.
Certain aspects of the manuscript need reinforcement with controls and some clarifications to warrant publication.
Points to address:
1. The authors should show that LPS contamination has no role in the phenotype. Use of TLR4 inhibitor (TAK242) or IRAK4 inhibitor can address this
2. Maturation of DC with CD83 should be supported with additional facs for CD80, CD86 and MHC-I/II. Costimulation and antigen presentation both are required for Tc response so these might be changed by the HPV protein treatment of DCs
3. Authors should include important control of untreated DC when comparing each HPV protein or combination. This is included in CD83 figure panel but missing in fig 3c,d and 4.
4. Cytokine IL12 and IL1b data are not convincingly explained. Why is there no difference in IL1b. Additionally, IL12 is only reduced by MAPK inh in p16, is there explanation for this? Why does E2+p16 decrease IL12 compared to p16 alone (compare PBS group black bars in fig 3d
5. if fig 4e, comparison should be done to untreated DC and not just between E2 v p16.
6. LPS treated DCs can be used as positive control in expt
7. fig 4e and 4f there is discrepancy: E2 increases IFNg in 4e (1:10) but in 4f E2 alone has no difference with p16 alone (compare no block groups). Also why only E2+p16 shows phenotype with HLA block needs clarifications.

---

## Round 0.2 · accepted · Accept

· Academic Editor

Accept

All critiques were adequately addressed, and the revised manuscript is acceptable now.

·

Basic reporting

No comment.

Experimental design

No comment.

Validity of the findings

No comment.

Additional comments

High-risk human papillomavirus (HPV) infection is known to be a necessary factor for cervical and anogenital malignancies. Cervical cancers account for over a quarter of a million deaths annually. Despite the availability of prophylactic vaccines, HPV infections remain extremely common worldwide. Furthermore, these vaccines are ineffective at clearing pre-existing infections and associated preinvasive lesions. As cervical dysplasia can regress spontaneously, a therapeutic HPV vaccine that boosts host immunity could have a significant impact on the morbidity and mortality associated with HPV. In this manuscript, Dr. Sunthamala and colleagues looked at the role of the HPV16 E2 and p16INK4a protein in stimulation of T lymphocytes, and wanted to provide a strategy for further in vivo study of stimulation of T lymphocytes for therapy of HPV-associated cancer. The study provides the conclusion, which showed that monocyte-derived dendritic cells pulsed with both proteins enhances specific response of both CD4+ and CD8+ T lymphocytes. The paper is significantly improved and all concerned raised by the reviewer have been addressed. I think it is suitable for publication at this point for this version of revised manuscript.